# On Tractable Computation of Expected Predictions

**Pasha Khosravi, YooJung Choi, Yitao Liang, Antonio Vergari, and Guy Van den Broeck**
Department of Computer Science
University of California, Los Angeles
{pashak,yjchoi,yliang,aver,guyvdb}@cs.ucla.edu

## Abstract

Computing *expected predictions* of discriminative models is a fundamental task in machine learning that appears in many interesting applications such as fairness, handling missing values, and data analysis. Unfortunately, computing expectations of a discriminative model with respect to a probability distribution defined by an arbitrary generative model has been proven to be hard in general. In fact, the task is intractable even for simple models such as logistic regression and a naive Bayes distribution. In this paper, we identify a pair of generative and discriminative models that enables tractable computation of expectations, as well as moments of any order, of the latter with respect to the former in case of regression. Specifically, we consider expressive probabilistic circuits with certain structural constraints that support tractable probabilistic inference. Moreover, we exploit the tractable computation of high-order moments to derive an algorithm to approximate the expectations for classification scenarios in which exact computations are intractable. Our framework to compute expected predictions allows for handling of missing data during prediction time in a principled and accurate way and enables reasoning about the behavior of discriminative models. We empirically show our algorithm to consistently outperform standard imputation techniques on a variety of datasets. Finally, we illustrate how our framework can be used for exploratory data analysis.

## 1 Introduction

Learning predictive models like regressors or classifiers from data has become a routine exercise in machine learning nowadays. Nevertheless, making predictions and reasoning about classifier behavior on unseen data is still a highly challenging task for many real-world applications. This is even more true when data is affected by uncertainty, e.g., in the case of noisy or missing observations.

A principled way to deal with this kind of uncertainty would be to probabilistically reason about the expected outcomes of a predictive model on a particular feature distribution. That is, to compute mathematical expectations of the predictive model w.r.t. a generative model representing the feature distribution. This is a common need that arises in many scenarios including dealing with missing data [20, 14], performing feature selection [37, 4, 7], handling sensor failure and resource scaling [12], seeking explanations [25, 21, 3] or determining how "fair" the learned predictor is [38, 39, 8].

While dealing with the above expectations is ubiquitous in machine learning, computing the expected predictions of an *arbitrary* discriminative models w.r.t. an *arbitrary* generative model is in general computationally intractable [14, 26]. As one would expect, the more expressive these models become, the harder it is to compute the expectations. More interestingly, even resorting to simpler discriminative models like logistic regression does not help reducing the complexity of such a task: computing the first moment of its predictions w.r.t. a naive Bayes model is known to be NP-hard [14].

In this work, we introduce a pair of expressive generative and discriminative models for regression, for which it is possible to compute not only expectations, but any moment efficiently. We leverage

recent advancements in probabilistic circuit representations. Specifically, we prove that generative and discriminative circuits enable computing the moments in time polynomial in the size of the circuits, when they are subject to some structural constraints which do not hinder their expressiveness.

Moreover, we demonstrate that for classification even the aforementioned structural constraints cannot guarantee computations in tractable time. However, efficiently approximating them becomes doable in polynomial time by leveraging our algorithm for the computations of arbitrary moments.

Lastly, we investigate applications of computing expectations. We first consider the challenging scenario of missing values at test time. There, we empirically demonstrate that computing expectations of a discriminative circuit w.r.t. a generative one is not only a more robust and accurate option than many imputation baselines for regression, but also for classification. In addition, we show how we can leverage this framework for exploratory data analysis to understand behavior of predictive models within different sub-populations.

## 2    Expectations and higher order moments of discriminative models

We use uppercase letters for random variables, e.g., $X$, and lowercase letters for their assignments e.g., $x$. Analogously, we denote sets of variables in bold uppercase, e.g., $\mathbf{X}$ and their assignments in bold lowercase, e.g., $\mathbf{x}$. The set of all possible values that $\mathbf{X}$ can take is denoted as $\mathcal{X}$.

Let $p$ be a probability distribution over $\mathbf{X}$ and $f : \mathcal{X} \to \mathbb{R}$ be a discriminative model, e.g., a regressor, that assigns a real value (outcome) to each complete input configuration $\mathbf{x} \in \mathcal{X}$ (features). The task of computing *the $k$-th moment* of $f$ with respect to the distribution $p$ is defined as:

$$M_k(f,p) \triangleq \mathbb{E}_{\mathbf{x} \sim p(\mathbf{x})} \left[ (f(\mathbf{x}))^k \right]. \tag{1}$$

Computing moments of arbitrary degree $k$ allows one to probabilistically reason about the outcomes of $f$. That is, it provides a description of the distribution of its predictions assuming $p$ as the data-generating distribution. For instance, we can compute the mean of $f$ w.r.t. $p$: $\mathbb{E}_p[f] = M_1(f,p)$ or reason about the dispersion (variance) of its outcomes: $\mathbb{VAR}_p(f) = M_2(f,p) - (M_1(f,p))^2$.

These computations can be a very useful tool to reason in a principled way about the behavior of $f$ in the presence of uncertainty, such as making predictions with missing feature values [14] or deciding a subset of $\mathbf{X}$ to observe [16, 37]. For example, given a partial assignment $\mathbf{x}^o$ to a subset $\mathbf{X}^o \subseteq \mathbf{X}$, the expected prediction of $f$ over the unobserved variables can be computed as $\mathbb{E}_{\mathbf{x} \sim p(\mathbf{x}|\mathbf{x}^o)} [f(\mathbf{x})]$, which is equivalent to $M_1(f, p(.|\mathbf{x}^o))$.

Unfortunately, computing arbitrary moments, and even just the expectation, of a discriminative model w.r.t. an arbitrary distribution is, in general, computationally hard. Under the restrictive assumptions that $p$ fully factorizes, i.e., $p(\mathbf{X}) = \prod_i p(X_i)$, and that $f$ is a simple linear model of the form $f(\mathbf{x}) = \sum_i \phi_i x_i$, computing expectations can be done in linear time. However, the task suddenly becomes NP-hard even for slightly more expressive models, for instance when $p$ is a naive Bayes distribution and $f$ is a logistic regression (a generalized linear model with a sigmoid activation function). See [14] for a detailed discussion.

In Section 4, we propose a pair of a generative and discriminative models that are highly expressive and yet still allow for polytime computation of exact moments and expectations of the latter w.r.t. the former. We first review the necessary background material in Section 3.

## 3    Generative and discriminative circuits

This section introduces the pair of circuit representations we choose as expressive generative and discriminative models. In both cases, we assume the input is discrete. We later establish under which conditions computing expected predictions becomes tractable.

**Logical circuits**    A *logical circuit* [11, 9] is a directed acyclic graph representing a logical formula where each node $n$ encodes a logical sub-formula, denoted as $[n]$. Each inner node in the graph is either an AND or an OR gate, and each leaf (input) node encodes a Boolean literal (e.g., $X$ or $\neg X$). We denote the set of child nodes of a gate $n$ as $\mathsf{ch}(n)$. An assignment $\mathbf{x}$ satisfies node $n$ if it satisfies the logical formula encoded by $n$, written $\mathbf{x} \models [n]$. Fig. 1 depicts some examples of

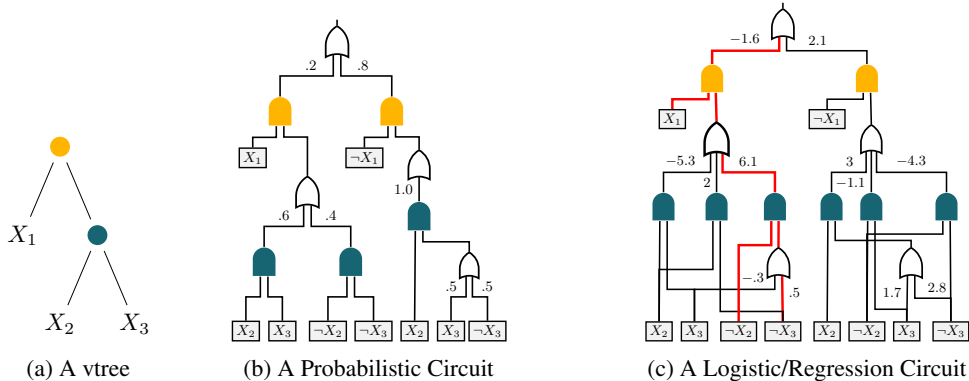

| (a) A vtree | (b) A Probabilistic Circuit | (c) A Logistic/Regression Circuit |

Figure 1: A vtree (a) over $\mathbf{X} = \{X_1, X_2, X_3\}$ and a generative and discriminative circuit pair (b, c) that conform to it. AND gates are colored as the vtree nodes they correspond to (blue and orange). For the discriminative circuit on the right, "hot wires" that form a path from input to output are colored red, for the given input configuration $\mathbf{x} = (X_1 = 1, X_2 = 0, X_3 = 0)$.

logical circuits. Several syntactic properties of circuits enable efficient logical and probabilistic reasoning over them [11, 29]. We now review those properties as they will be pivotal for our efficient computations of expectations and high-order moments in Section 4.

**Syntactic Properties** A circuit is said to be *decomposable* if for every AND gate its inputs depend on disjoint sets of variables. For notational simplicity, we will assume decomposable AND gates to have two inputs, denoted L(eft) and R(ight) children, depending on variables $\mathbf{X}^\mathsf{L}$ and $\mathbf{X}^\mathsf{R}$ respectively. In addition, a circuit satisfies *structured decomposability* if each of its AND gates decomposes according to a *vtree*, a binary tree structure whose leaves are the circuit variables. That is, the L (resp. R) child of an AND gate depends on variables that appear on the left (resp. right) branch of its corresponding vtree node. Fig. 1 shows a vtree and visually maps its nodes to the AND gates of two example circuits. A circuit is *smooth* if for an OR gate all its children depend on the same set of variables [32]. Lastly, a circuit is *deterministic* if, for any input, at most one child of every OR node has a non-zero output. For example, Fig. 1c highlights in red the wires that are true, and that form a path from the root to the leaves, given input $\mathbf{x} = (X_1 = 1, X_2 = 0, X_3 = 0)$. Note that every OR gate in Fig. 1c has at most one hot input wire, because of the determinism property.

**Generative probabilistic circuits** A *probabilistic circuit* (PC) is characterized by its logical circuit structure and parameters $\theta$ that are assigned to the inputs of each OR gate.

Intuitively, each PC node $n$ recursively defines a distribution $p_n$ over a subset of the variables $\mathbf{X}$ appearing in the sub-circuit rooted at it. More precisely:

$$p_n(\mathbf{x}) = \begin{cases} \mathbb{1}_n(\mathbf{x}) & \text{if } n \text{ is a leaf,} \\ p_\mathsf{L}(\mathbf{x}^\mathsf{L}) \cdot p_\mathsf{R}(\mathbf{x}^\mathsf{R}) & \text{if } n \text{ is an AND gate,} \\ \sum_{i \in \mathsf{ch}(n)} \theta_i p_i(\mathbf{x}) & \text{if } n \text{ is an OR gate.} \end{cases} \tag{2}$$

Here, $\mathbb{1}_n(\mathbf{x}) \triangleq \mathbb{1}\{\mathbf{x} \models [n]\}$ indicates whether the leaf $n$ is satisfied by input $\mathbf{x}$. Moreover, $\mathbf{x}^\mathsf{L}$ and $\mathbf{x}^\mathsf{R}$ indicate the subsets of configuration $\mathbf{x}$ restricted to the decomposition defined by an AND gate over its L (resp. R) child. As such, an AND gate of a PC represents a factorization over independent sets of variables, whereas an OR gate defines a mixture model. Unless otherwise noted, in this paper we adopt PCs that satisfy *structured decomposability* and *smoothness* as our generative circuit.

PCs allow for the exact computation of the probability of complete and partial configurations (that is, marginalization) in time linear in the size of the circuit. A well-known example of PCs is the probabilistic sentential decision diagram (PSDD) [15].[1] They have been successfully employed as state-of-the-art density estimators not only for unstructured [19] but also for structured feature spaces [5, 30, 31]. Other types of PCs include sum-product networks (SPNs) and cutset networks, yet those representations are typically decomposable but not structured decomposable [23, 24].

**Discriminative circuits** For the discriminative model $f$, we adopt and extend the semantics of logistic circuits (LCs): discriminative circuits recently introduced for classification [18]. An LC is defined by a decomposable, *smooth* and *deterministic* logical circuit with parameters $\phi$ on inputs to OR gates. Moreover, we will work with LCs that are *structured decomposable*, which is a restriction already supported by their learning algorithms [18]. An LC acts as a classifier on top of a rich set of non-linear features, extracted by its logical circuit structure. Specifically, an LC assigns an *embedding* representation $h(\mathbf{x})$ to each input example $\mathbf{x}$. Each feature $h(\mathbf{x})_k$ in the embedding is associated with one input $k$ of one of the OR gates in the circuit (and thus also with one parameter $\phi_k$). It corresponds to a logical formula that can be readily extracted from the logical circuit structure.

Classification is performed on this new feature representation by applying a sigmoid non-linearity: $f^{\mathsf{LC}}(\mathbf{x}) \triangleq 1/(1 + e^{-\sum_k \phi_k h(\mathbf{x})_k})$, and similar to logistic regression it is amenable to convex parameter optimization. Alternatively, one can fully characterize an LC by recursively defining the output of each node $m$. We use $g_m(\mathbf{x})$ to define output of node $m$ given $\mathbf{x}$. It can be computed as:

$$g_m(\mathbf{x}) = \begin{cases} 0 & \text{if } m \text{ is a leaf,} \\ g_{\mathsf{L}}(\mathbf{x}^{\mathsf{L}}) + g_{\mathsf{R}}(\mathbf{x}^{\mathsf{R}}) & \text{if } m \text{ is an AND gate,} \\ \sum_{j \in \mathsf{ch}(m)} \mathbb{1}_j(\mathbf{x})(\phi_j + g_j(\mathbf{x})) & \text{if } m \text{ is an OR gate.} \end{cases} \quad (3)$$

Again, $\mathbb{1}_j(\mathbf{x})$ is an indicator for $\mathbf{x} \models [j]$, effectively using the determinism property of LCs to select which input to pass through. Then classification is done by applying a sigmoid function to the output of the circuit root $r$: $f^{\mathsf{LC}}(\mathbf{x}) = 1/(1 + e^{-g_r(\mathbf{x})})$. The increased expressive power of LCs w.r.t. simple linear regressors lies in the rich representations $h(\mathbf{x})$ they learn, which in turn rely on the underlying circuit structure as a powerful feature extractor [34, 33].

LCs have been introduced for classification and were shown to outperform larger neural networks [18]. We also leverage them for regression, that is, we are interested in computing the expectations of the output of the root node $g_r(\mathbf{x})$ w.r.t. a generative model $p$. We call an LC when no sigmoid function is applied to $g_r(\mathbf{x})$ a *regression circuit* (RC). As we will show in the next section, we are able to exactly compute *any moment* of an RC $g$ w.r.t. an LC $p$, that is, $M_k(g, p)$, in time polynomial in the size of the circuits, if $p$ and $g$ share the same vtree.

## 4 Computing expectations and moments for circuit pairs

We now introduce our main result, which leads to efficient algorithms for tractable Expectation and Moment Computation of Circuit pairs ($\mathsf{EC}_2$ and $\mathsf{MC}_2$) in which the discriminative model is an RC and the generative model is a PC, and where both circuits are structured decomposable *sharing the same vtree*. Recall that we also assumed all circuits to be smooth, and the RC to be deterministic.

**Theorem 1.** *Let $n$ and $m$ be root nodes of a PC and an RC with the same vtree over $\mathbf{X}$. Let $s_n$ and $s_m$ be their respective number of edges. Then, the $k^{th}$ moment of $g_m$ w.r.t. the distribution encoded by $p_n$, that is, $M_k(g_m, p_n)$, can be computed exactly in time $O(k^2 s_n s_m)$.*[2]

Moreover, this complexity is attained by the $\mathsf{MC}_2$ algorithm, which we describe in the next section. We then investigate how this result can be generalized to arbitrary circuit pairs and how restrictive the structural requirements are. In fact, we demonstrate how computing expectations and moments for circuit pairs *not* sharing a vtree is #P-hard. Furthermore, we address the hardness of computing expectations for an LC w.r.t. a PC–due to the introduction of the sigmoid function over $g$–by approximating it through the tractable computation of moments with the $\mathsf{MC}_2$ algorithm.

### 4.1 $\mathsf{EC}_2$: Expectations of regression circuits

Intuitively, the computation of expectations becomes tractable because we can "break it down" to the leaves of the PC and RC, where it reduces to trivial computations. Indeed, the two circuits sharing the same vtree is the property that enables a polynomial time recursive decomposition, because it ensures that pairs of nodes considered by the algorithm depend on exactly the same set of variables.

**Algorithm 1** $\text{EC}_2(n, m)$                    ▷ Cache recursive calls to achieve polynomial complexity

---

**Require:** A PC node $n$ and an RC node $m$
    **if** $m$ is Leaf **then return** $0$
    **else if** $n$ is Leaf **then**
        **if** $[n] \models [m_\mathsf{L}]$ **then return** $\phi_{m_\mathsf{L}}$
        **if** $[n] \models [m_\mathsf{R}]$ **then return** $\phi_{m_\mathsf{R}}$
    **else if** $n,m$ are OR **then return** $\sum_{i \in \mathsf{ch}(n)} \theta_i \sum_{j \in \mathsf{ch}(m)} (\text{EC}_2(i, j) + \phi_j \text{PR}(i, j))$
    **else if** $n,m$ are AND **then return** $\text{PR}(n_\mathsf{L}, m_\mathsf{L})\, \text{EC}_2(n_\mathsf{R}, m_\mathsf{R}) + \text{PR}(n_\mathsf{R}, m_\mathsf{R})\, \text{EC}_2(n_\mathsf{L}, m_\mathsf{L})$

---

We will now show how this computation recursively decomposes over pairs of OR and AND gates, starting from the roots of the PC $p$ and RC $g$. We refer the reader to the Appendix for detailed proofs of all Propositions and Theorems in this section. Without loss of generality, we assume that the roots of both $p$ and $g$ are OR gates, and that circuit nodes alternate between AND and OR gates layerwise.

**Proposition 1.** *Let $n$ and $m$ be OR gates of a PC and an RC, respectively. Then the expectation of the regressor $g_m$ w.r.t. distribution $p_n$ is:*

$$M_1(g_m, p_n) = \sum_{i \in \mathsf{ch}(n)} \theta_i \sum_{j \in \mathsf{ch}(m)} (M_1(\mathbb{1}_j \cdot g_j, p_i) + \phi_j M_1(\mathbb{1}_j, p_i)).$$

The above proposition illustrates how the expectation of an OR gate of an RC w.r.t. an OR gate in the PC is a weighted sum of the expectations of the child nodes. The number of smaller expectations to be computed is quadratic in the number of children. More specifically, one now has to compute expectations of two different functions w.r.t. the children of PC $n$.

First, $M_1(\mathbb{1}_j, p_i)$ is the expectation of the indicator function associated to the $j$-th child of $m$ (see Eq. 3) w.r.t. the $i$-th child node of $n$. Intuitively, this translates to the probability of the logical formula $[j]$ being satisfied according to the distribution encoded by $p_i$. Fortunately, this can be computed efficiently, in quadratic time, linear in the size of both circuits as already demonstrated in [5].

On the other hand, computing the other expectation term $M_1(\mathbb{1}_j g_j, p_i)$ requires a novel algorithm tailored to RCs and PCs. We next show how to further decompose this expectation from AND gates to their OR children.

**Proposition 2.** *Let $n$ and $m$ be AND gates of a PC and an RC, respectively. Let $n_\mathsf{L}$ and $n_\mathsf{R}$ (resp. $m_\mathsf{L}$ and $m_\mathsf{R}$) be the left and right children of $n$ (resp. $m$). Then the expectation of function $(\mathbb{1}_m \cdot g_m)$ w.r.t. distribution $p_n$ is:*

$$M_1(\mathbb{1}_m \cdot g_m, p_n) = M_1(\mathbb{1}_{m_\mathsf{L}}, p_{n_\mathsf{L}}) M_1(g_{m_\mathsf{R}}, p_{n_\mathsf{R}}) + M_1(\mathbb{1}_{m_\mathsf{R}}, p_{n_\mathsf{R}}) M_1(g_{m_\mathsf{L}}, p_{n_\mathsf{L}}).$$

We are again left with the task of computing expectations of the RC node indicator functions, i.e., $M_1(\mathbb{1}_{m_\mathsf{L}}, p_{n_\mathsf{L}})$ and $M_1(\mathbb{1}_{m_\mathsf{R}}, p_{n_\mathsf{R}})$, which can also be done by exploiting the algorithm in [5]. Furthermore, note that the other expectation terms ($M_1(g_{m_\mathsf{L}}, p_{n_\mathsf{L}})$ and $M_1(g_{m_\mathsf{R}}, p_{n_\mathsf{R}})$) can readily be computed using Proposition 1, since they concern pairs of OR nodes.

We briefly highlight how determinism in the regression circuit plays a crucial role in enabling this computation. In fact, OR gates being deterministic ensures that the otherwise non-decomposable product of indicator functions $\mathbb{1}_m \cdot \mathbb{1}_k$, where $m$ is a parent OR gate of an AND gate $k$, results to be equal to $\mathbb{1}_k$. We refer the readers to Appendix A.3 for a detailed discussion.

Recursively, one is guaranteed to reach pairs of leaf nodes in the RC and PC, for which the respective expectations can be computed in $\mathcal{O}(1)$ by checking if their associated Boolean indicators agree, and by noting that $g_m(\mathbf{x}) = 0$ if $m$ is a leaf (see Eq. 3). Putting it all together, we obtain the recursive procedure shown in Algorithm 1. Here, $\text{PR}(n, m)$ refer to the algorithm to compute $M_1(\mathbb{1}_m, p_n)$ in [5]. As the algorithm computes expectations in a bottom-up fashion, the intermediate computations can be cached to avoid evaluating the same pair of nodes more than once, and therefore keeping the complexity as stated by our Theorem 1.

## 4.2 MC$_2$: Moments of regression circuits

Our algorithmic solution goes beyond the tractable computation of the sole expectation of an RC. Indeed, any arbitrary order moment of $g_m$ can be computed w.r.t. $p_n$, still in polynomial time. We call this algorithm MC$_2$ and we delineate its main routines with the following Propositions:[3]

**Proposition 3.** *Let $n$ and $m$ be OR gates of a PC and an RC, respectively. Then the $k$-th moment of the regressor $g_m$ w.r.t. distribution $p_n$ is:*

$$M_k(g_m, p_n) = \sum_{i \in \mathsf{ch}(n)} \theta_i \sum_{j \in \mathsf{ch}(m)} \sum_{l=0}^{k} \binom{k}{l} \phi_j^{k-l} M_l(\mathbb{1}_j \cdot g_j, p_i).$$

**Proposition 4.** *Let $n$ and $m$ be AND gates of a PC and an RC, respectively. Let $n_\mathsf{L}$ and $n_\mathsf{R}$ (resp. $m_\mathsf{L}$ and $m_\mathsf{R}$) be the left and right children of $n$ (resp. $m$). Then the $k$-th moment of function $(\mathbb{1}_m g_m)$ w.r.t. distribution $p_n$ is:*

$$M_k(\mathbb{1}_m \cdot g_m, p_n) = \sum_{l=0}^{k} \binom{k}{l} M_l(\mathbb{1}_{m_\mathsf{L}} \cdot g_{m_\mathsf{L}}, p_{n_\mathsf{L}}) M_{k-l}(\mathbb{1}_{m_\mathsf{R}} \cdot g_{m_\mathsf{R}}, p_{n_\mathsf{R}})$$

Analogous to computing simple expectations, by recursively and alternatively applying Propositions 3 and 4, we arrive at the moments of the leaves at both circuits, while gradually reducing the order $k$ of the involved moments.

Furthermore, the lower-order moments in Proposition 4 that decompose to L and R children, e.g., $M_l(\mathbb{1}_{m_\mathsf{L}} \cdot g_{m_\mathsf{L}}, p_{n_\mathsf{L}})$, can be computed by noting that they reduce to:

$$M_k(\mathbb{1}_m \cdot g_m, p_n) = \begin{cases} M_1(\mathbb{1}_m, p_n) & \text{if } k = 0, \\ M_k(g_m, p_n) & \text{otherwise.} \end{cases} \tag{4}$$

Note again that these computations are made possible by the interplay of determinism of $g$ and shared vtrees between $p$ and $g$. From the former it follows that a sum over OR gate children reduces to a single child value. The latter ensures that the AND gates in $p$ and $g$ decompose in the same way, thereby enabling efficient computations.

Given this, a natural question arises: "*If we do not require a PC $p$ and a RC $g$ to have the same vtree structure, is computing $M_k(g, p)$ still tractable?*". Unfortunately, this is not the case, as we demonstrate in the following theorem.

**Theorem 2.** *Computing any moment of an RC $g_m$ w.r.t. a PC distribution $p_n$ where both have arbitrary vtrees is #P-hard.*

At a high level, we can reduce #SAT, a well known #P-complete problem on CNF sentences, to the moment computation problem. Given a choice of different vtrees, we can construct an RC and a PC in time polynomial in the size of the CNF formula such that its #SAT value can be computed using the expectation of the RC w.r.t. the PC. We refer to Appendix A.3 for more details.

So far, we have focused our analysis to RCs, the analogous of LCs for regression. One would hope that the efficient computations of EC$_2$ could be carried on to LCs to compute the expected predictions of classifiers. However, the application of the sigmoid function $\sigma$ on the regressor $g$, *even when $g$ shares the same vtree as $p$*, makes the problem intractable, as our next Theorem shows.

**Theorem 3.** *Taking the expectation of an LC ($\sigma \circ g_m$) w.r.t. a PC distribution $p_n$ is NP-hard even if $n$ and $m$ share the same vtree.*

This follows from a recent result that taking the expectation of a logistic regression w.r.t. a naive Bayes distribution is NP-hard [14]; see Appendix A.4 for a detailed proof.

## 4.3 Approximating expectations of classifiers

Theorem 3 leaves us with no hope of computing exact expected predictions in a tractable way even for pairs of generative PCs and discriminative LCs conforming to the same vtree. Nevertheless, we can leverage the ability to efficiently compute the moments of the RC $g_m$ to efficiently approximate

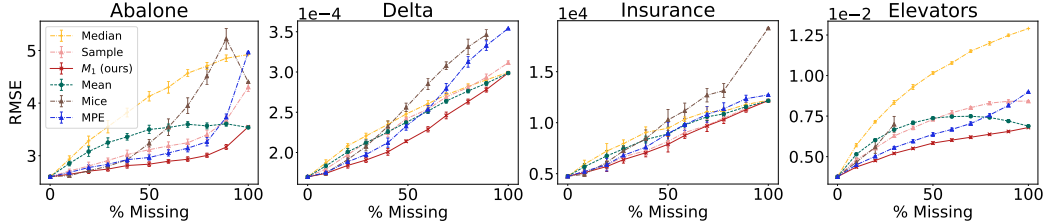

Figure 2: Evaluating $EC_2$ for predictions under different percentages of missing features (x-axis) over four real-world *regression* datasets in terms of the RMSE (y-axis) of the predictions of $g((\mathbf{x}^m, \mathbf{x}^o))$. Overall, exactly computing the expected predictions via $EC_2$ outperforms simple imputation schemes like median and mean as well as more sophisticated ones like MICE [1] or computing the MPE configuration with the PC $p$. Detailed dataset statistics can be found in Appendix B.

the expectation of $\gamma \circ g_m$, with $\gamma$ being any differentiable non-linear function, including sigmoid $\sigma$. Using a Taylor series approximation around point $\alpha$ we define the following $d$-order approximation:

$$T_d(\gamma \circ g_m, p_n) \triangleq \sum_{k=0}^{d} \frac{\gamma^{(k)}(\alpha)}{k!} M_k(g_m - \alpha, p_n)$$

See Appendix A.5, for a detailed derivation and more intuition behind this approximation.

## 5 Expected prediction in action

In this section, we empirically evaluate the usefulness and effectiveness of computing the expected predictions of our discriminative circuits with respect to generative ones.[4] First, we tackle the challenging task of making predictions in the presence of missing values at test time, for both regression and classification.[5] Second, we show how our framework can be used to reasoning about the behavior of predictive models. We employ it in the context of exploratory data analysis, to check for biases in the predictive models, or to search for interesting patterns in the predictions associated with sub-populations in the data distribution.

### 5.1 Reasoning with missing values: an application

Traditionally, prediction with missing values has been addressed by imputation, which substitutes missing values with presumably reasonable alternatives such as the mean or median, estimated from training data [28]. These imputation methods are typically heuristic and model-agnostic [20]. To overcome this, the notion of expected predictions has recently been proposed in [14] as a probabilistically principled and model-aware way to deal with missing values. Formally, we want to compute

$$\mathbb{E}_{\mathbf{x}^m \sim p(\mathbf{x}^m|\mathbf{x}^o)} \left[ f(\mathbf{x}^m \mathbf{x}^o) \right] \tag{5}$$

where $\mathbf{x}^m$ (resp. $\mathbf{x}^o$) denotes the configuration of a sample $\mathbf{x}$ that is missing (resp. observed) at test time. In the case of regression, we can exactly compute Eq. 5 for a pair of generative and discriminative circuits sharing the same vtree by our proposed algorithm, after observing that

$$\mathbb{E}_{\mathbf{x}^m \sim p(\mathbf{x}^m|\mathbf{x}^o)} \left[ f(\mathbf{x}^m \mathbf{x}^o) \right] = \frac{1}{p(\mathbf{x}^o)} \mathbb{E}_{\mathbf{x}^m \sim p(\mathbf{x}^m, \mathbf{x}^o)} \left[ f(\mathbf{x}^m \mathbf{x}^o) \right] \tag{6}$$

where $p(\mathbf{x}^m, \mathbf{x}^o)$ is the unnormalized distribution encoded by the generative circuit *configured* for evidence $\mathbf{x}^o$. That is, the sub-circuits depending on the variables in $\mathbf{X}^o$ have been fixed according to the input $\mathbf{x}^o$. This transformation, and computing any marginal $p(\mathbf{x}^o)$, can be done efficiently in time linear in the size of the PC [10].

To demonstrate the generality of our method, we construct a 6-dataset testing suite, four of which are common regression benchmarks from several domains [13], and the rest are classification on MNIST

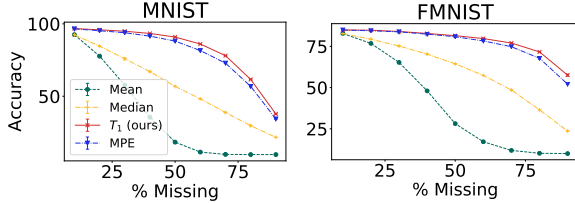

Figure 3: Evaluating the first-order Taylor approximation $T_1(\sigma \circ g_m, p_n)$ of the expected predictions of a *classifier* for missing value imputation for different percentages of missing features (x-axis) in terms of the accuracy (y-axis).

and FASHION datasets [36, 35]. We compare our method with classical imputation techniques such as standard mean and median imputation, and more sophisticated (and computationally intensive) imputation techniques such as multiple imputations by chained equations (MICE) [1]. Moreover, we adopt a natural and strong baseline: imputing the missing values by the most probable explanation (MPE) [10], computed by probabilistic reasoning on the generative circuit $p$. Note that the MPE inference acts as an imputation: it returns the mode of the input feature distribution, while $EC_2$ would convey a more global statistic of the distribution of the outputs of such a predictive model.

To enforce that the discriminative-generative pair of circuits share the same vtree, we first generate a fixed random and balanced vtree and use it to guide the respective parameter and structure learning algorithms of our circuits. In our experiments we adopt PSDDs [15] for the generative circuits. PSDDs are a subset of PCs, since they also satisfy determinism. Although we do not require determinism of generative circuits for moment computation, we use PSDDs due to the availability of their learning algorithms.

On image data, however, we exploit the already learned and publicly available LC structure in [18], which scores 99.4% accuracy on MNIST, being competitive to much larger deep models. We learn a PSDD with the same vtree. For RCs, we adapt the parameter and structure learning of LCs [18], substituting the logistic regression objective with a ridge regression during optimization. For structure learning of both LCs and RCs, we considered up to 100 iterates while monitoring the loss on a held out set. For PSDDs we employ the parameter and structure learning of [19] with default parameters and run it up to 1000 iterates until no significant improvement is seen on a held out set.

Figure 2 shows our method outperforming other regression baselines. This can be explained by the fact that it computes the exact expectation while other techniques make restrictive assumptions to approximate the expectation. Mean and median imputations effectively assume that the features are independent; MICE[6] assumes a fixed dependence formula between the features; and, as already stated, MPE only considers the highest probability term in the expansion of the expectation.

Additionally, as we see in Figure 3, our approximation method for predicted classification, using just the first-order expansion $T_1(\gamma \circ g_m, p_n)$, is able to outperform the predictions of the other competitors. This suggests that our method is effective in approximating the true expected values.

These experiments agree with the observations from [14] that, given missing data, probabilistically reasoning about the outcome of a classifier by taking expectations can generally outperform imputation techniques. Our advantage clearly comes from the PSDD learning a better density estimation of the data distribution, instead of having fixed prior assumptions about the features. An additional demonstration of this fact comes from the excellent performance of MPE on both datasets. Again, this can be credited to the PSDD learning a good distribution on the features.

## 5.2 Reasoning about predictive models for exploratory data analysis

We now showcase an example of how our framework can be utilized for exploratory data analysis while reasoning about the behavior of a given predictive model. Suppose an insurance company has hired us to analyze both their data and the predictions of their regression model. To simulate this scenario, we use the RC and PC circuits that were learned on the real-world Insurance dataset in the previous section (see Figure 2). This dataset lists the yearly health insurance cost of individuals living in the US with features such as age, smoking habits, and location. Our task is to examine the behavior of the predictions, such as whether they are biased by some sensitive attributes or whether there exist interesting patterns across sub-populations of the data.

We might start by asking: *"how different are the insurance costs between smokers and non smokers?"* which can be easily computed as

$$M_1(f, \; p(. \mid Smoker)) - M_1(f, \; p(. \mid Non\ Smoker)) = 31,355 - 8,741 = 22,614 \qquad (7)$$

by applying the same conditioning as in Equations 5 and 6. We can also ask: *"is the predictive model biased by gender?"* To answer this question, it would be interesting to compute:

$$M_1(f, \; p(. \mid Female)) - M_1(f, \; p(. \mid Male)) = 14,170 - 13,196 = 974 \qquad (8)$$

As expected, being a smoker affects the health insurance costs much more than being male or female. If it were the opposite, we would conclude that the model may be unfair or misbehaving.

In addition to examining the effect of a single feature, we may study the model in a smaller sub-population, by conditioning the distribution on multiple features. For instance, suppose the insurance company is interested in expanding and as part of their marketing plan wants to know the effect of an individual's region, e.g., southeast (SE) and southwest (SW), for the sub-population of female (F) smokers (S) with one child (C). By computing the following quantities, we can discover that the difference in their average insurance cost is relevant, but much more relevant is the difference in their standard deviations, indicating a significantly different treatment of this population between regions:

$$\mathbb{E}_{p_{\mathsf{SE}}}[f] = M_1(f, p(. \mid \mathsf{F}, \mathsf{S}, \mathsf{C}, \mathsf{SE})) = 30,974, \; \mathbb{STD}_{p_{\mathsf{SE}}}[f] = \sqrt{M_2(.) - (M_1(.))^2} = 11,229 \quad (9)$$

$$\mathbb{E}_{p_{\mathsf{SW}}}[f] = M_1(f, p(. \mid \mathsf{F}, \mathsf{S}, \mathsf{C}, \mathsf{SW})) = 27,250, \; \mathbb{STD}_{p_{\mathsf{SW}}}[f] = \sqrt{M_2(.) - (M_1(.))^2} = 7,717 \quad (10)$$

However, one may ask why we do not estimate these values directly from the dataset. The main issue in doing so is that as we condition on more features, fewer if not zero matching samples are present in the data. For example, only 4 and 3 samples match the criterion asked by the last two queries. Furthermore, it is not uncommon for the data to be unavailable due to sensitivity or privacy concerns, and only the models are available. For instance, two insurance agencies in different regions might want to partner without sharing their data yet.

The expected prediction framework with probabilistic circuits allows us to efficiently compute these queries with interesting applications in explainability and fairness. We leave the more rigorous exploration of their applications for future work.

## 6   Related Work

Using expected prediction to handle missing values was introduced in Khosravi et al. [14]; given a logistic regression model, they learned a conforming Naive bayes model and then computed expected prediction only using the learned naive bayes model. In contrast, we are taking the expected prediction using two distinct models. Moreover, probabilistic circuits are much more expressive models. Imputations are a common way to handle missing features and are a well-studied topic. For more detail and a history of the techniques we refer the reader to Buuren [2], Little and Rubin [20].

Probabilistic circuits enable a wide range of tractable operations. Given the two circuits, our expected prediction algorithm operated on the pairs of children of the nodes in the two circuits corresponding to the same vtree node and hence had a quadratic run-time. There are other applications that operate on similar pairs of nodes such as: multiplying the distribution of two PSDDs [29], computing the probability of a logical formula [6], and computing KL divergence [17].

## 7   Conclusion

In this paper we investigated under which model assumptions it is tractable to compute expectations of certain discriminative models. We proved how, for regression, pairing a discriminative circuit with a generative one sharing the same vtree structure allows to compute not only expectations but also arbitrary high-order moments in poly-time. Furthermore, we characterized when the task is otherwise hard, e.g., for classification, when a non-decomposable, non-linear function is introduced. At the same time, we devised for this scenario an approximate computation that leverages the aforementioned efficient computation of the moments of regressors. Finally, we showcased how the expected prediction framework can help a data analyst to reason about the predictive model's behavior under different sub-populations. This opens up several interesting research venues, from applications like reasoning about missing values, to perform feature selection, to scenarios where exact and approximate computations of expected predictions can be combined.

## Acknowledgements

This work is partially supported by NSF grants #IIS-1633857, #CCF-1837129, DARPA XAI grant #N66001-17-2-4032, NEC Research, and gifts from Intel and Facebook Research.

## Footnotes

[1] PSDDs by definition also satisfy determinism, but we do not require this property for computing moments.

[2] This is a loose upper bound since the algorithm only looks at a small subset of pairs of edges in the circuits. A tighter bound would be $O(k^2 \sum_v s_v t_v)$ where $v$ ranges over vtree nodes and $s_v$ (resp. $t_v$) counts the number of edges going into the nodes of the PC (resp. RC) that can be attributed to the vtree node $v$.

[3] The algorithm MC$_2$ can easily be derived from EC$_2$ in Algorithm 1, using the equations in this section.

[4]Our implementation of the algorithm and experiments are available at https://github.com/UCLA-StarAI/mc2.

[5]In case of classification, we use the Taylor expansion approximation we discussed in Section 4.3.

[6]On the elevator dataset, we reported MICE result only until 30% missing as the imputation method is computationally heavy and required more than 10hr to complete.

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
