[Supplementary Material 1 · circuit_expect_neurips19_supplementary.pdf]

# Supplement "On Tractable Computation of Expected Predictions"

## A  Proofs

### A.1  Proofs of Propositions 1 and 3

We will first prove Proposition 3, from which Proposition 1 directly follows. For a PC OR node $n$ and RC OR node $m$,

$$M_k(g_m, p_n) = \mathop{\mathbb{E}}_{\mathbf{x} \sim p_n(\mathbf{x})} \left[ g_m^k(\mathbf{x}) \right]$$

$$= \mathop{\mathbb{E}}_{\mathbf{x} \sim p_n(\mathbf{x})} \left[ \left( \sum_{j \in \mathsf{ch}(m)} \mathbb{1}_j(\mathbf{x})(g_j(\mathbf{x}) + \phi_j) \right)^k \right]$$

$$= \mathop{\mathbb{E}}_{\mathbf{x} \sim p_n(\mathbf{x})} \sum_{j \in \mathsf{ch}(m)} \left[ (\mathbb{1}_j(\mathbf{x})(g_j(\mathbf{x}) + \phi_j))^k \right] \tag{11}$$

$$= \mathop{\mathbb{E}}_{\mathbf{x} \sim p_n(\mathbf{x})} \sum_{j \in \mathsf{ch}(m)} \sum_{l=0}^{k} \binom{k}{l} g_j^l(\mathbf{x}) \phi_j^{k-l} \mathbb{1}_j(\mathbf{x})$$

$$= \sum_{\mathbf{x}} p_n(\mathbf{x}) \sum_{j \in \mathsf{ch}(m)} \sum_{l=0}^{k} \binom{k}{l} g_j^l(\mathbf{x}) \phi_j^{k-l} \mathbb{1}_j(\mathbf{x})$$

$$= \sum_{\mathbf{x}} \sum_{i \in \mathsf{ch}(n)} \theta_i p_i(\mathbf{x}) \sum_{j \in \mathsf{ch}(m)} \sum_{l=0}^{k} \binom{k}{l} g_j^l(\mathbf{x}) \phi_j^{k-l} \mathbb{1}_j(\mathbf{x})$$

$$= \sum_{i \in \mathsf{ch}(n)} \theta_i \sum_{j \in \mathsf{ch}(m)} \sum_{l=0}^{k} \binom{k}{l} \phi_j^{k-l} \sum_{\mathbf{x}} p_i(\mathbf{x}) g_j^l(\mathbf{x}) \mathbb{1}_j(\mathbf{x})$$

$$= \sum_{i \in \mathsf{ch}(n)} \theta_i \sum_{j \in \mathsf{ch}(m)} \sum_{l=0}^{k} \binom{k}{l} \phi_j^{k-l} \mathop{\mathbb{E}}_{\mathbf{x} \sim p_i(\mathbf{x})} [\mathbb{1}_j(\mathbf{x}) g_j^l(\mathbf{x})]$$

$$= \sum_{i \in \mathsf{ch}(n)} \theta_i \sum_{j \in \mathsf{ch}(m)} \sum_{l=0}^{k} \binom{k}{l} \phi_j^{k-l} M_l(\mathbb{1}_j \cdot g_j, p_i). \tag{12}$$

Equation 11 follows from determinism of RCs as at most one $j$ will have a non-zero $\mathbb{1}_j(\mathbf{x})$. In Equation 12, note that we denote, with slight abuse of notation, $M_0(\mathbb{1}_j \cdot g_j, p_i) = \mathbb{E}_{\mathbf{x} \sim p_i(\mathbf{x})}[\mathbb{1}_j(\mathbf{x})] = M_1(\mathbb{1}_j, p_i)$. This concludes the proof of Proposition 3.

We obtain Proposition 1 by applying above result with $k = 1$:

$$M_1(g_m, p_n) = \sum_{i \in \mathsf{ch}(n)} \theta_i \sum_{j \in \mathsf{ch}(m)} \sum_{l=0}^{1} \binom{1}{l} \phi_j^{1-l} M_l(\mathbb{1}_j \cdot g_j, p_i)$$

$$= \sum_{i \in \mathsf{ch}(n)} \theta_i \sum_{j \in \mathsf{ch}(m)} (\phi_j M_0(\mathbb{1}_j \cdot g_j, p_i) + M_1(\mathbb{1}_j \cdot g_j, p_i))$$

$$= \sum_{i \in \mathsf{ch}(n)} \theta_i \sum_{j \in \mathsf{ch}(m)} (\phi_j M_1(\mathbb{1}_j, p_i) + M_1(\mathbb{1}_j \cdot g_j, p_i)).$$

### A.2  Proofs of Proposition 2 and 4

Again, we will first prove Proposition 4. For a PC AND node $n$ and RC AND node $m$,

$$M_k(\mathbb{1}_m g_m, p_n) = \mathop{\mathbb{E}}_{\mathbf{x} \sim p_n(\mathbf{x})} \left[ \mathbb{1}_m(\mathbf{x}) g_m^k(\mathbf{x}) \right]$$

$$= \underset{\mathbf{x} \sim p_n(\mathbf{x})}{\mathbb{E}} \left[ \mathbb{1}_m(\mathbf{x}) \left( g_{m_\mathsf{L}}(\mathbf{x}^\mathsf{L}) + g_{m_\mathsf{R}}(\mathbf{x}^\mathsf{R}) \right)^k \right]$$

$$= \sum_{\mathbf{x}^\mathsf{L}, \mathbf{x}^\mathsf{R}} p_{n_\mathsf{L}}(\mathbf{x}^\mathsf{L}) p_{n_\mathsf{R}}(\mathbf{x}^\mathsf{R}) \mathbb{1}_m(\mathbf{x}) \left( g_{m_\mathsf{L}}(\mathbf{x}^\mathsf{L}) + g_{m_\mathsf{R}}(\mathbf{x}^\mathsf{R}) \right)^k$$

$$= \sum_{\mathbf{x}^\mathsf{L}, \mathbf{x}^\mathsf{R}} p_{n_\mathsf{L}}(\mathbf{x}^\mathsf{L}) p_{n_\mathsf{R}}(\mathbf{x}^\mathsf{R}) \mathbb{1}_{m_\mathsf{L}}(\mathbf{x}^\mathsf{L}) \mathbb{1}_{m_\mathsf{R}}(\mathbf{x}^\mathsf{R}) \sum_{l=0}^{k} \binom{k}{l} g_{m_\mathsf{L}}^l(\mathbf{x}^\mathsf{L}) g_{m_\mathsf{R}}^{k-l}(\mathbf{x}^\mathsf{R}) \qquad (13)$$

$$= \sum_{l=0}^{k} \binom{k}{l} \left( \sum_{\mathbf{x}^\mathsf{L}} p_{n_\mathsf{L}}(\mathbf{x}^\mathsf{L}) \mathbb{1}_{m_\mathsf{L}}(\mathbf{x}^\mathsf{L}) g_{m_\mathsf{L}}^l(\mathbf{x}^\mathsf{L}) \right) \left( \sum_{\mathbf{x}^\mathsf{R}} p_{n_\mathsf{R}}(\mathbf{x}^\mathsf{R}) \mathbb{1}_{m_\mathsf{R}}(\mathbf{x}^\mathsf{R}) g_{m_\mathsf{R}}^{k-l}(\mathbf{x}^\mathsf{R}) \right)$$

$$= \sum_{l=0}^{k} \binom{k}{l} \underset{\mathbf{x}^\mathsf{L} \sim p_{n_\mathsf{L}}(\mathbf{x}^\mathsf{L})}{\mathbb{E}} \left[ \mathbb{1}_{m_\mathsf{L}}(\mathbf{x}^\mathsf{L}) g_{m_\mathsf{L}}^l(\mathbf{x}^\mathsf{L}) \right] \underset{\mathbf{x}^\mathsf{R} \sim p_{n_\mathsf{R}}(\mathbf{x}^\mathsf{R})}{\mathbb{E}} \left[ \mathbb{1}_{m_\mathsf{R}}(\mathbf{x}^\mathsf{R}) g_{m_\mathsf{R}}^{k-l}(\mathbf{x}^\mathsf{R}) \right]$$

$$= \sum_{l=0}^{k} \binom{k}{l} M_l(\mathbb{1}_{m_\mathsf{L}} \cdot g_{m_\mathsf{L}}, p_{n_\mathsf{L}}) M_{k-l}(\mathbb{1}_{m_\mathsf{R}} \cdot g_{m_\mathsf{R}}, p_{n_\mathsf{R}}).$$

Equation 13 follows from decomposability: $\mathbb{1}_m(\mathbf{x}) = \mathbb{1}\{\mathbf{x} \models [m]\} = \mathbb{1}\{\mathbf{x} \models [m_\mathsf{L} \wedge m_\mathsf{R}]\} = \mathbb{1}\{\mathbf{x}^\mathsf{L} \models [m_\mathsf{L}]\} \mathbb{1}\{\mathbf{x}^\mathsf{R} \models [m_\mathsf{R}]\} = \mathbb{1}_{m_\mathsf{L}}(\mathbf{x}^\mathsf{L}) \mathbb{1}_{m_\mathsf{R}}(\mathbf{x}^\mathsf{R})$. This concludes the proof of Proposition 4.

We obtain Proposition 2 by combining above result at $k = 1$ with Equation 4:

$$M_1(\mathbb{1}_m \cdot g_m, p_n)$$

$$= \sum_{l=0}^{1} \binom{1}{l} M_l(\mathbb{1}_{m_\mathsf{L}} \cdot g_{m_\mathsf{L}}, p_{n_\mathsf{L}}) M_{1-l}(\mathbb{1}_{m_\mathsf{R}} \cdot g_{m_\mathsf{R}}, p_{n_\mathsf{R}})$$

$$= M_0(\mathbb{1}_{m_\mathsf{L}} \cdot g_{m_\mathsf{L}}, p_{n_\mathsf{L}}) M_1(\mathbb{1}_{m_\mathsf{R}} \cdot g_{m_\mathsf{R}}, p_{n_\mathsf{R}}) + M_1(\mathbb{1}_{m_\mathsf{L}} \cdot g_{m_\mathsf{L}}, p_{n_\mathsf{L}}) M_0(\mathbb{1}_{m_\mathsf{R}} \cdot g_{m_\mathsf{R}}, p_{n_\mathsf{R}})$$

$$= M_1(\mathbb{1}_{m_\mathsf{L}}, p_{n_\mathsf{L}}) M_1(g_{m_\mathsf{R}}, p_{n_\mathsf{R}}) + M_1(\mathbb{1}_{m_\mathsf{R}}, p_{n_\mathsf{R}}) M_1(g_{m_\mathsf{L}}, p_{n_\mathsf{L}}).$$

## A.3 Proof of Theorem 2

The proof is by reduction from the model counting problem (#SAT) which is known to be #P-hard.

Given a CNF formula $\alpha$, let us construct $\beta$ and $\gamma$ as follows. For every variable $X_i$ appearing in clause $\alpha_j$, introduce an auxiliary variable $X_{ij}$. Then:

$$\beta \equiv \bigwedge_i \left( X_{i1} \Leftrightarrow \cdots \Leftrightarrow X_{ij} \Leftrightarrow \cdots \Leftrightarrow X_{im} \right),$$

$$\gamma \equiv \bigwedge_j \bigvee_i l_\alpha(X_{ij}).$$

Here, $l_\alpha(X_{ij})$ denotes the literal of $X_i$ (i.e., $X_i$ or $\neg X_i$) in clause $\alpha_j$. Thus, $\gamma$ is the same CNF formula as $\alpha$, except that a variable in $\alpha$ appears as several different copies in $\gamma$. The formula $\beta$ ensures that the copied variables are all equivalent. Thus, the model count of $\alpha$ must equal the model count of $\beta \wedge \gamma$.

Consider a right-linear vtree in which variables appear in the following order: $X_{11}, X_{12}, \ldots, X_{1j}, \ldots, X_{ij}, \ldots$. The PC sub-circuit involving copies of variable $X_i$ has exactly two model and size that is linear in the number of copies. There are as many such sub-circuits as there are variables in the original formula $\alpha$, each of which can be chained together directly to obtain $\beta$. The key insight in doing so is that sub-circuits corresponding to different variables $X_i$ are independent of one another. Then, we can construct a PC circuit structure whose logical formula represents $\beta$ in polytime. In a single top down pass, we can parameterize the PC $p_n$ such that it represents a uniform distribution: each model is assigned a probability of $1/2^n$.

Next, consider a right-linear vtree with the variables appearing in the following order: $X_{11}, X_{21}, \ldots, X_{n1}, \ldots, X_{ij}, \ldots$. Then, we can construct a logical circuit that represents $\gamma$ in polynomial time, as each variable appears exactly once in the formula. That is, each clause $\alpha_j$ will have a PC sub-circuit with linear size (in the number of literals appearing in the clause), and the size

of their conjunction $\alpha$ will simply be the sum of the sizes of such sub-circuits. We can parameterize it as a regression circuit $g_m$ by assigning 0 to all inputs to OR gates and adding a single OR gate on top of the root node with a weight 1. Then this regression circuit outputs 1 if and only if the input assignment satisfies $\gamma$.

Then the expectation of regression circuit $g_m$ w.r.t. PC $p_n$ (which does not share the same vtree) can be used to compute the model count of $\alpha$ as follows:

$$M_1(g_m, p_n) = \mathop{\mathbb{E}}_{\mathbf{x} \sim p_n(\mathbf{x})}[g_m(\mathbf{x})] = \sum_{\mathbf{x}} p_n(\mathbf{x}) g_m(\mathbf{x}) = \sum_{\mathbf{x}} \frac{1}{2^n} \mathbb{1}[\mathbf{x} \models \beta] \mathbb{1}[\mathbf{x} \models \gamma]$$

$$= \frac{1}{2^n} \sum_{\mathbf{x}} \mathbb{1}[\mathbf{x} \models \beta \wedge \gamma] = \frac{1}{2^n} \mathrm{MC}(\beta \wedge \gamma) = \frac{1}{2^n} \mathrm{MC}(\alpha)$$

Thus, #SAT can be reduced to the problem of computing expectations of a regression circuit w.r.t. a PC that does not share the same vtree. □

## A.4 Proof of Theorem 3

The proof is by reduction from computing expectation of a logistic regression w.r.t. a naive Bayes distribution, which was shown to be NP-hard.

Given a naive Bayes distribution $P(\mathbf{X}, C)$, we can build a PC $p_n$ that represents the same distribution in polynomial time by employing a right-linear vtree in which the class variable appears at the top, followed by the features. Because the feature distribution conditioned on the class variable is fully factorized, the PC sub-circuits corresponding to $P(\mathbf{X}|C)$ and $P(\mathbf{X}|\neg C)$ will each have size that is linear in the number of features.

Moreover, given a logistic regression model $f(\mathbf{x}) = \sigma(w(\mathbf{x}))$, we can build a corresponding logistic circuit $\sigma \circ g_m$ in polytime using the same vtree as the PC described previously. Specifically, each non-leaf node $v$ in the vtree corresponds to an AND gate, and for each its child we add an OR gate with paramter 0 (to keep the structure of alternating between AND and OR gates), recursively building the circuit. The leaf node for each variable $X$ become an OR gate with 2 children $X$ and $\neg X$, with parameters $w_i$ and 0, respectively. The leaf nodes involving the class variable $C$ will simply have weights 0. As shown in [18], logistic circuits become equivalent to logistic regression on the feature embedding space, define by the structure of the circuit, as well as the "raw" features. With this parameterization, we ensure that the extra features introduced by the logistic circuit structure always have weight 0, so overall the circuit becomes equivalent to the original logistic regression. That is, $w(\mathbf{x}) = g_m(C, \mathbf{x}) = g_m(\neg C, \mathbf{x})$ for all assignments $\mathbf{x}$.

Figure 4 gives an example of the construction of the circuits using a given vtree, logistic regression, and a naive Bayes model. The logistic regression model is defined as $f(\mathbf{x}) = \sum_i \mathbf{x}_i w_i$, and for the naive Bayes model parameters are $\theta_c = P(c)$, $\theta_{x_i|c} = P(\mathbf{x}_i \mid c)$, and $\theta_{x_i|\neg c} = P(\mathbf{x}_i \mid \neg c)$. Other values can be easily computed using the complement rule, for example $\theta_{\neg x_i|c} = 1 - \theta_{x_i|c}$. Finally, the naive Bayes distribution is now defined as: $P(\mathbf{x}, C) = \theta_C \prod_i \theta_{x_i|C}$.

The expectation of such logistic circuit $\sigma \circ g_m$ w.r.t. PC $p_n$ is equal to the expectation of original logistic regression $f$ w.r.t. naive Bayes $P$ as the following:

$$M_1(\sigma \circ g_m, p_n) = \mathop{\mathbb{E}}_{c\mathbf{x} \sim p_n(c\mathbf{x})}[\sigma(g_m(c\mathbf{x}))] = \mathop{\mathbb{E}}_{c\mathbf{x} \sim P(c\mathbf{x})}[f(\mathbf{x})] = \mathop{\mathbb{E}}_{\mathbf{x} \sim P(\mathbf{x})}[f(\mathbf{x})].$$

□

## A.5 Approximating expected prediction of classifiers

In this section, we provide more intuition on how we derived our approximation method for the case of classification. As mentioned in the main text, we define the following $d$-order approximation:

$$T_d(\gamma \circ g_m, p_n) \triangleq \sum_{k=0}^{d} \frac{\gamma^{(k)}(\alpha)}{k!} M_k(g_m - \alpha, p_n)$$

We can use $T_d(\gamma \circ g_m, p_n)$ as an approximation to $M_1(\gamma \circ g_m, p_n)$ because:

$$M_1(\gamma \circ g_m, p_n) = \mathbb{E}_{\mathbf{x} \sim p_n(\mathbf{x})}\left[\gamma(g_m(\mathbf{x}))\right] = \mathbb{E}_{\mathbf{x} \sim p_n(\mathbf{x})} \sum_{i=0}^{\infty} \frac{\gamma^{(i)}(\alpha)}{i!} \big(g_m(\mathbf{x}) - \alpha\big)^i$$

(a) A vtree       (b) Logistic Regression as a Logistic Circuit conforming to the vtree

(c) Naive Bayes as a PC conforming to the vtree

Figure 4: A vtree (a) over $\mathbf{X} = \{X_1, X_2, X_3\}$ and corresponding circuits that are respectively equivalent to a given Logistic Regression model with parameters $w_0, w_1, w_2, w_3$, and a Naive Bayes model with parameters $\theta_c$, $\theta_{x_i|c}$, $\theta_{x_i|\neg c}$.

$$\approx \sum_{i=0}^{d} \frac{\gamma^{(i)}(\alpha)}{i!} \mathbb{E}_{\mathbf{x} \sim p_n(\mathbf{x})} \left(g_m(\mathbf{x}) - \alpha\right)^i = T_d(\gamma \circ g_m, p_n)$$

For example, given a PC with root $n$ and a logistic circuit with root $m$ and sigmoid activation, the Taylor series around point $\alpha = 0$ and $d = 5$ gives us:

$$M_1(\gamma \circ g_m, p_n) \approx T_5(\gamma \circ g_m, p_n) = \frac{1}{2} + \frac{M_1(g_m, p_n)}{4} - \frac{M_3(g_m, p_n)}{48} + \frac{M_5(g_m, p_n)}{480}$$

In general, we would like to expand the Taylor series around a point that converges quickly. In our case, we employ $\alpha \approx M_1(g_m, p_n)$. All these Taylor expansion terms can be computed efficiently, as long as taking the derivatives of our non-linearity can be done efficiently at point $\alpha$.

Table 1: Statistics as number of train, validation and test samples and features (after discretization) for the datasets employed in the regression (top half) and classification (bottom half) experiments.

| DATASET | TRAIN | VALID | TEST | FEATURES |
|---|---|---|---|---|
| ABALONE | 2923 | 584 | 670 | 71 |
| DELTA-AIRLOINS | 4990 | 998 | 1141 | 55 |
| ELEVATORS | 11619 | 2323 | 2657 | 182 |
| INSURANCE | 936 | 187 | 215 | 36 |
| MNIST | 48000 | 12000 | 10000 | 784 |
| FASHION | 48000 | 12000 | 10000 | 784 |

## B  Datasets

We employed the following datasets for our empirical evaluation, taken from the UCI Machine Learning repository and other regression [13] or classification suites.

**Description of the datasets**    ABALONE[7] [22] contains several physical measurements on abalone specimens used to predict their age. DELTA-AIRLOINS collects mechanical measurements for the task of controlling the ailerons of a F16 aircraft while the task is to predict the variation of the action on the ailerons. ELEVATORS comprises measurements also concerned with the task of controlling a F16 aircraft (different from DELTA-AIRLOINS), although the target variable here refers on controlling on the elevators of the aircraft. In INSURANCE[8] one wants to predict individual medical costs billed by health insurance given several personal data of a patient. MNIST [9] comprises gray-scale handwritten digit images used for multi-class classification. FASHION [10] is a 10-class image classification challenge concerning fashion apparel items.

**Preprocessing steps**    We preserve for all dataset their train and test splits if present in their respective repositories, or create a new test set comprising 20% of the whole data. Moreover we reserve a 10% portion of the training set as validation data used to monitor (parameter and/or structure) learning of our models and perform early-stopping.

We perform discretization of the continuous features in the regression datasets as follows. We first try to automatically detect the optimal number of (irregular) bins through adaptive binning by employing a penalized likelihood scheme as in [27]. If the number of the bins found in this way exceeds ten, we employ an equal-width binning scheme capping the bin number to ten, instead. Once the data is discrete, we encode them as binary through the common one-hot encoding, to accommodate the requirements of the PSDD learner we employed [19].

For image data, we binarize each sample by considering each pixel in it to be 1 if its original value exceed the mean value of that pixel as computed on the training set.

Statistics for all the datasets after preprocessing can be found in Table 1.

**Runtime of the algorithms**    In Table 2, we report the runtimes for our method versus MICE and expectations approximated via Monte Carlo simulations, by sampling the generative model and evaluating on the discriminative model. As we see, the speed advantage of our algorithm becomes more clear on larger datasets. The runtime of the Monte Carlo approximation depends on number of samples and the size of the generative circuit. The runtime of MICE also depends on missing percentage of features and increases notably as more features go missing. For this reason, and by observing that MICE was providing worse predictions than our algorithm, we stopped MICE experiments early at 30% missing for the ELEVATORS dataset.

Table 2: Statistics on the runtime of our algorithm versus MICE and the Monte Carlo Sampling algorithm. The reported times for prediction times are for one configuration of the experiment. As we tried 10 different missingness percentages and repeated each 10 times, the total time of experiment is 100 times the value in the table. Learning time refers to learning the generative circuit and is done only once.

| | Time (seconds) | | | |
| | **ours** (learning) | **ours** (prediction) | **MICE** | **MC** |
| --- | --- | --- | --- | --- |
| ABALONE | 82 | 20 | 43 | 117 |
| DELTA | 53 | 24 | 27 | 126 |
| INSURANCE | 40 | 13 | 11 | 20 |
| ELEVATORS | 2105 | 31 | 364 | 994 |

**Computing Infrastructure**   The experiments were run on a combination of a server with 40 CPU cores and 500 GB of RAM, and a laptop with 6 CPU cores and 16 GB of RAM. The server was mainly utilized for learning the circuits, albeit not using all the memory, and to parallelize different runs of the missing value experiments. No GPUs were used for the experiments as probabilistic circuit libraries do not support them yet.

To report the runtimes in Table 2, we did a separate run of each method on the same machine (the laptop) for fair comparison of the runtimes.

## Footnotes

[7]https://archive.ics.uci.edu/ml/datasets/abalone

[8]https://www.kaggle.com/mirichoi0218/insurance

[9]http://yann.lecun.com/exdb/mnist/

[10]https://github.com/zalandoresearch/fashion-mnist


[Supplementary Material 2]

## Supplements "On Tractable Computation of Expected Predictions"

## A Proofs

### A.1 Proofs of Propositions 1 and 3

We will first prove Proposition 3, from which Proposition 1 directly follows. For a PSDD OR node $n$ and RC OR node $m$,

$$M_k(g_m, p_n) = \mathop{\mathbb{E}}_{\mathbf{x} \sim p_n(\mathbf{x})} \left[ g_m^k(\mathbf{x}) \right]$$

$$= \mathop{\mathbb{E}}_{\mathbf{x} \sim p_n(\mathbf{x})} \left[ \left( \sum_{j \in \mathsf{ch}(m)} \mathbb{1}_j(\mathbf{x})(g_j(\mathbf{x}) + \phi_j) \right)^k \right]$$

$$= \mathop{\mathbb{E}}_{\mathbf{x} \sim p_n(\mathbf{x})} \sum_{j \in \mathsf{ch}(m)} \left[ (\mathbb{1}_j(\mathbf{x})(g_j(\mathbf{x}) + \phi_j))^k \right] \qquad (11)$$

$$= \mathop{\mathbb{E}}_{\mathbf{x} \sim p_n(\mathbf{x})} \sum_{j \in \mathsf{ch}(m)} \sum_{l=0}^{k} \binom{k}{l} g_j^l(\mathbf{x}) \phi_j^{k-l} \mathbb{1}_j(\mathbf{x})$$

$$= \sum_{\mathbf{x}} p_n(\mathbf{x}) \sum_{j \in \mathsf{ch}(m)} \sum_{l=0}^{k} \binom{k}{l} g_j^l(\mathbf{x}) \phi_j^{k-l} \mathbb{1}_j(\mathbf{x})$$

$$= \sum_{\mathbf{x}} \sum_{i \in \mathsf{ch}(n)} \theta_i p_i(\mathbf{x}) \sum_{j \in \mathsf{ch}(m)} \sum_{l=0}^{k} \binom{k}{l} g_j^l(\mathbf{x}) \phi_j^{k-l} \mathbb{1}_j(\mathbf{x})$$

$$= \sum_{i \in \mathsf{ch}(n)} \theta_i \sum_{j \in \mathsf{ch}(m)} \sum_{l=0}^{k} \binom{k}{l} \phi_j^{k-l} \sum_{\mathbf{x}} p_i(\mathbf{x}) g_j^l(\mathbf{x}) \mathbb{1}_j(\mathbf{x})$$

$$= \sum_{i \in \mathsf{ch}(n)} \theta_i \sum_{j \in \mathsf{ch}(m)} \sum_{l=0}^{k} \binom{k}{l} \phi_j^{k-l} \mathop{\mathbb{E}}_{\mathbf{x} \sim p_i(\mathbf{x})} [\mathbb{1}_j(\mathbf{x}) g_j^l(\mathbf{x})]$$

$$= \sum_{i \in \mathsf{ch}(n)} \theta_i \sum_{j \in \mathsf{ch}(m)} \sum_{l=0}^{k} \binom{k}{l} \phi_j^{k-l} M_l(\mathbb{1}_j \cdot g_j, p_i). \qquad (12)$$

Equation 11 follows from determinism as at most one $j$ will have a non-zero $\mathbb{1}_j(\mathbf{x})$. In Equation 12, note that we denote, with slight abuse of notation, $M_0(\mathbb{1}_j \cdot g_j, p_i) = \mathbb{E}_{\mathbf{x} \sim p_i(\mathbf{x})}[\mathbb{1}_j(\mathbf{x})] = M_1(\mathbb{1}_j, p_i)$. This concludes the proof of Proposition 3.

We obtain Proposition 1 by applying above result with $k = 1$:

$$M_1(g_m, p_n) = \sum_{i \in \mathsf{ch}(n)} \theta_i \sum_{j \in \mathsf{ch}(m)} \sum_{l=0}^{1} \binom{1}{l} \phi_j^{1-l} M_l(\mathbb{1}_j \cdot g_j, p_i)$$

$$= \sum_{i \in \mathsf{ch}(n)} \theta_i \sum_{j \in \mathsf{ch}(m)} (\phi_j M_0(\mathbb{1}_j \cdot g_j, p_i) + M_1(\mathbb{1}_j \cdot g_j, p_i))$$

$$= \sum_{i \in \mathsf{ch}(n)} \theta_i \sum_{j \in \mathsf{ch}(m)} (\phi_j M_1(\mathbb{1}_j, p_i) + M_1(\mathbb{1}_j \cdot g_j, p_i)).$$

### A.2 Proofs of Proposition 2 and 4

Again, we will first prove Proposition 4. For a PSDD AND node $n$ and RC AND node $m$,

$$M_k(\mathbb{1}_m g_m, p_n) = \mathop{\mathbb{E}}_{\mathbf{x} \sim p_n(\mathbf{x})} \left[ \mathbb{1}_m(\mathbf{x}) g_m^k(\mathbf{x}) \right]$$

$$= \mathop{\mathbb{E}}_{\mathbf{x} \sim p_n(\mathbf{x})} \left[ \mathbb{1}_m(\mathbf{x}) \left( g_{m_\mathsf{L}}(\mathbf{x}^\mathsf{L}) + g_{m_\mathsf{R}}(\mathbf{x}^\mathsf{R}) \right)^k \right]$$

$$= \sum_{\mathbf{x}^\mathsf{L}, \mathbf{x}^\mathsf{R}} p_{n_\mathsf{L}}(\mathbf{x}^\mathsf{L}) p_{n_\mathsf{R}}(\mathbf{x}^\mathsf{R}) \mathbb{1}_m(\mathbf{x}) \left( g_{m_\mathsf{L}}(\mathbf{x}^\mathsf{L}) + g_{m_\mathsf{R}}(\mathbf{x}^\mathsf{R}) \right)^k$$

$$= \sum_{\mathbf{x}^\mathsf{L}, \mathbf{x}^\mathsf{R}} p_{n_\mathsf{L}}(\mathbf{x}^\mathsf{L}) p_{n_\mathsf{R}}(\mathbf{x}^\mathsf{R}) \mathbb{1}_{m_\mathsf{L}}(\mathbf{x}^\mathsf{L}) \mathbb{1}_{m_\mathsf{R}}(\mathbf{x}^\mathsf{R}) \sum_{l=0}^{k} \binom{k}{l} g_{m_\mathsf{L}}^l(\mathbf{x}^\mathsf{L}) g_{m_\mathsf{R}}^{k-l}(\mathbf{x}^\mathsf{R}) \qquad (13)$$

$$= \sum_{l=0}^{k} \binom{k}{l} \left( \sum_{\mathbf{x}^\mathsf{L}} p_{n_\mathsf{L}}(\mathbf{x}^\mathsf{L}) \mathbb{1}_{m_\mathsf{L}}(\mathbf{x}^\mathsf{L}) g_{m_\mathsf{L}}^l(\mathbf{x}^\mathsf{L}) \right) \left( \sum_{\mathbf{x}^\mathsf{R}} p_{n_\mathsf{R}}(\mathbf{x}^\mathsf{R}) \mathbb{1}_{m_\mathsf{R}}(\mathbf{x}^\mathsf{R}) g_{m_\mathsf{R}}^{k-l}(\mathbf{x}^\mathsf{R}) \right)$$

$$= \sum_{l=0}^{k} \binom{k}{l} \mathop{\mathbb{E}}_{\mathbf{x}^\mathsf{L} \sim p_{n_\mathsf{L}}(\mathbf{x}^\mathsf{L})} \left[ \mathbb{1}_{m_\mathsf{L}}(\mathbf{x}^\mathsf{L}) g_{m_\mathsf{L}}^l(\mathbf{x}^\mathsf{L}) \right] \mathop{\mathbb{E}}_{\mathbf{x}^\mathsf{R} \sim p_{n_\mathsf{R}}(\mathbf{x}^\mathsf{R})} \left[ \mathbb{1}_{m_\mathsf{R}}(\mathbf{x}^\mathsf{R}) g_{m_\mathsf{R}}^{k-l}(\mathbf{x}^\mathsf{R}) \right]$$

$$= \sum_{l=0}^{k} \binom{k}{l} M_l(\mathbb{1}_{m_\mathsf{L}} \cdot g_{m_\mathsf{L}}, p_{n_\mathsf{L}}) M_{k-l}(\mathbb{1}_{m_\mathsf{R}} \cdot g_{m_\mathsf{R}}, p_{n_\mathsf{R}}).$$

Equation 13 follows from decomposability: $\mathbb{1}_m(\mathbf{x}) = \mathbb{1}\{\mathbf{x} \models [m]\} = \mathbb{1}\{\mathbf{x} \models [m_\mathsf{L} \wedge m_\mathsf{R}]\} = \mathbb{1}\{\mathbf{x}^\mathsf{L} \models [m_\mathsf{L}]\} \mathbb{1}\{\mathbf{x}^\mathsf{R} \models [m_\mathsf{R}]\} = \mathbb{1}_{m_\mathsf{L}}(\mathbf{x}^\mathsf{L}) \mathbb{1}_{m_\mathsf{R}}(\mathbf{x}^\mathsf{R})$. This concludes the proof of Proposition 4.

We obtain Proposition 2 by combining above result at $k = 1$ with Equation 4:

$$M_1(\mathbb{1}_m \cdot g_m, p_n)$$

$$= \sum_{l=0}^{1} \binom{1}{l} M_l(\mathbb{1}_{m_\mathsf{L}} \cdot g_{m_\mathsf{L}}, p_{n_\mathsf{L}}) M_{1-l}(\mathbb{1}_{m_\mathsf{R}} \cdot g_{m_\mathsf{R}}, p_{n_\mathsf{R}})$$

$$= M_0(\mathbb{1}_{m_\mathsf{L}} \cdot g_{m_\mathsf{L}}, p_{n_\mathsf{L}}) M_1(\mathbb{1}_{m_\mathsf{R}} \cdot g_{m_\mathsf{R}}, p_{n_\mathsf{R}}) + M_1(\mathbb{1}_{m_\mathsf{L}} \cdot g_{m_\mathsf{L}}, p_{n_\mathsf{L}}) M_0(\mathbb{1}_{m_\mathsf{R}} \cdot g_{m_\mathsf{R}}, p_{n_\mathsf{R}})$$

$$= M_1(\mathbb{1}_{m_\mathsf{L}}, p_{n_\mathsf{L}}) M_1(g_{m_\mathsf{R}}, p_{n_\mathsf{R}}) + M_1(\mathbb{1}_{m_\mathsf{R}}, p_{n_\mathsf{R}}) M_1(g_{m_\mathsf{L}}, p_{n_\mathsf{L}}).$$

### A.3 Proof of Theorem 2

The proof is by reduction from the model counting problem (#SAT) which is known to be #P-hard.

Given a CNF formula $\alpha$, let us construct $\beta$ and $\gamma$ as follows. For every variable $X_i$ appearing in clause $\alpha_j$, introduce an auxiliary variable $X_{ij}$. Then:

$$\beta \equiv \bigwedge_i \left( X_{i1} \Leftrightarrow \cdots \Leftrightarrow X_{ij} \Leftrightarrow \cdots \Leftrightarrow X_{im} \right),$$

$$\gamma \equiv \bigwedge_j \bigvee_i l_\alpha(X_{ij}).$$

Here, $l_\alpha(X_{ij})$ denotes the literal of $X_i$ (i.e., $X_i$ or $\neg X_i$) in clause $\alpha_j$. Thus, $\gamma$ is the same CNF formula as $\alpha$, except that a variable in $\alpha$ appears as several different copies in $\gamma$. The formula $\beta$ ensures that the copied variables are all equivalent. Thus, the model count of $\alpha$ must equal the model count of $\beta \wedge \gamma$.

Consider a right-linear vtree in which variables appear in the following order: $X_{11}, X_{12}, \ldots, X_{1j}, \ldots, X_{ij}, \ldots$. The PSDD sub-circuit involving copies of variable $X_i$ has exactly two model and size that is linear in the number of copies. There are as many such sub-circuits as there are variables in the original formula $\alpha$, each of which can be chained together directly to obtain $\beta$. The key insight in doing so is that sub-circuits corresponding to different variables $X_i$ are independent of one another. Then, we can construct a PSDD circuit structure whose logical formula represents $\beta$ in polytime. In a single top down pass, we can parameterize the PSDD $p_n$ such that it represents a uniform distribution: each model is assigned a probability of $1/2^n$.

Next, consider a right-linear vtree with the variables appearing in the following order: $X_{11}, X_{21}, \ldots, X_{n1}, \ldots, X_{ij}, \ldots$. Then, we can construct a logical circuit that represents $\gamma$ in polynomial time, as each variable appears exactly once in the formula. That is, each clause $\alpha_j$ will have a PSDD sub-circuit with linear size (in the number of literals appearing in the clause), and

the size of their conjunction $\alpha$ will simply be the sum of the sizes of such sub-circuits. We can parameterize it as a regression circuit $g_m$ by assigning 0 to all inputs to OR gates and adding a single OR gate on top of the root node with a weight 1. Then this regression circuit outputs 1 if and only if the input assignment satisfies $\gamma$.

Then the expectation of regression circuit $g_m$ w.r.t. PSDD $p_n$ (which does not share the same vtree) can be used to compute the model count of $\alpha$ as follows:

$$M_1(g_m, p_n) = \mathop{\mathbb{E}}_{\mathbf{x} \sim p_n(\mathbf{x})} [g_m(\mathbf{x})] = \sum_{\mathbf{x}} p_n(\mathbf{x}) g_m(\mathbf{x}) = \sum_{\mathbf{x}} \frac{1}{2^n} \mathbb{1}[\mathbf{x} \models \beta] \mathbb{1}[\mathbf{x} \models \gamma]$$

$$= \frac{1}{2^n} \sum_{\mathbf{x}} \mathbb{1}[\mathbf{x} \models \beta \wedge \gamma] = \frac{1}{2^n} \mathrm{MC}(\beta \wedge \gamma) = \frac{1}{2^n} \mathrm{MC}(\alpha)$$

Thus, #SAT can be reduced to the problem of computing expectations of a regression circuit w.r.t. a PSDD that does not share the same vtree. $\square$

## A.4 Proof of Theorem 3

The proof is by reduction from computing expectation of a logistic regression w.r.t. a naive Bayes distribution, which was shown to be NP-hard.

Given a naive Bayes distribution $P(\mathbf{X}, C)$, we can build a PSDD $p_n$ that represents the same distribution in polynomial time by employing a right-linear vtree in which the class variable appears at the top, followed by the features. Because the feature distribution conditioned on the class variable is fully factorized, the PSDD sub-circuits corresponding to $P(\mathbf{X}|C)$ and $P(\mathbf{X}|\neg C)$ will each have size that is linear in the number of features.

Moreover, given a logistic regression model $f(\mathbf{x}) = \sigma(w(\mathbf{x}))$, we can build a corresponding logistic circuit $\sigma \circ g_m$ in polytime using the same vtree as the PSDD described previously. Specifically, each non-leaf node $v$ in the vtree corresponds to an AND gate, and for each its child we add an OR gate with paramter 0 (to keep the structure of alternating between AND and OR gates), recursively building the circuit. The leaf node for each variable $X$ become an OR gate with 2 children $X$ and $\neg X$, with parameters $w_i$ and 0, respectively. The leaf nodes involving the class variable $C$ will simply have weights 0. As shown in [15], logistic circuits become equivalent to logistic regression on the feature embedding space, define by the structure of the circuit, as well as the "raw" features. With this parametrization, we ensure that the extra features introduced by the logistic circuit structure will alway have weight 0, so overall the circuit becomes equivalent to the original logistic regression. That is, $w(\mathbf{x}) = g_m(C, \mathbf{x}) = g_m(\neg C, \mathbf{x})$ for all assignments $\mathbf{x}$.

Figure 4 gives an example of the construction of the circuits using a given vtree, logistic regression, and a naive Bayes model. The logistic regression model is defined as $f(\mathbf{x}) = \sum_i \mathbf{x}_i w_i$, and for the naive Bayes model parameters are $\theta_c = P(c)$, $\theta_{x_i|c} = P(\mathbf{x}_i \mid c)$, and $\theta_{x_i|\neg c} = P(\mathbf{x}_i \mid \neg c)$. Other values can be easily computed using the complement rule, for example $\theta_{\neg x_i|c} = 1 - \theta_{x_i|c}$. Finally, the naive Bayes distribution is now defined as: $P(\mathbf{x}, C) = \theta_C \prod_i \theta_{x_i|C}$.

The expectation of such logistic circuit $\sigma \circ g_m$ w.r.t. PSDD $p_n$ is equal to the expectation of original logistic regression $f$ w.r.t. naive Bayes $P$ as the following:

$$M_1(\sigma \circ g_m, p_n) = \mathop{\mathbb{E}}_{c\mathbf{x} \sim p_n(c\mathbf{x})} [\sigma(g_m(c\mathbf{x}))] = \mathop{\mathbb{E}}_{c\mathbf{x} \sim P(c\mathbf{x})} [f(\mathbf{x})] = \mathop{\mathbb{E}}_{\mathbf{x} \sim P(\mathbf{x})} [f(\mathbf{x})].$$

$\square$

## A.5 Approximating expected prediction of classifiers

In this section, we provide more intuition on how we derived our approximation method for the case of classification. As mentioned in the main text, we define the following $d$-order approximation:

$$T_d(\gamma \circ g_m, p_n) \triangleq \sum_{k=0}^{d} \frac{\gamma^{(k)}(\alpha)}{k!} M_k(g_m - \alpha, p_n)$$

We can use $T_d(\gamma \circ g_m, p_n)$ as an approximation to $M_1(\gamma \circ g_m, p_n)$ because:

$$M_1(\gamma \circ g_m, p_n) = \mathbb{E}_{\mathbf{x} \sim p_n(\mathbf{x})} \left[ \gamma(g_m(\mathbf{x})) \right] = \mathbb{E}_{\mathbf{x} \sim p_n(\mathbf{x})} \sum_{i=0}^{\infty} \frac{\gamma^{(i)}(\alpha)}{i!} (g_m(\mathbf{x}) - \alpha)^i$$

(a) A vtree

(b) Logistic Regression as a Logistic Circuit conforming to the vtree

(c) Naive Bayes as a PSDD conforming to the vtree

Figure 4: A vtree (a) over $\mathbf{X} = \{X_1, X_2, X_3\}$ and corresponding circuits that are respectively equivalent to a given Logistic Regression model with parameters $w_0, w_1, w_2, w_3$, and a Naive Bayes model with parameters $\theta_c$, $\theta_{x_i|c}$, $\theta_{x_i|\neg c}$.

$$\approx \sum_{i=0}^{d} \frac{\gamma^{(i)}(\alpha)}{i!} \mathbb{E}_{\mathbf{x} \sim p_n(\mathbf{x})} \left(g_m(\mathbf{x}) - \alpha\right)^i = T_d(\gamma \circ g_m, p_n)$$

For example, given a PSDD with root $n$ and a logistic circuit with root $m$ and sigmoid activation, the Taylor series around point $\alpha = 0$ and $d = 5$ gives us:

$$M_1(\gamma \circ g_m, p_n) \approx T_5(\gamma \circ g_m, p_n) = \frac{1}{2} + \frac{M_1(g_m, p_n)}{4} - \frac{M_3(g_m, p_n)}{48} + \frac{M_5(g_m, p_n)}{480}$$

In general, we would like to expand the Taylor series around a point that converges quickly. In our case, we employ $\alpha \approx M_1(g_m, p_n)$. All these Taylor expansion terms can be computed efficiently, as long as taking the derivatives of our non-linearity can be done efficiently at point $\alpha$.

Table 1: Statistics as number of train, validation and test samples and features (after discretization) for the datasets employed in the regression (top half) and classification (bottom half) experiments.

| DATASET | TRAIN | VALID | TEST | FEATURES |
|---|---|---|---|---|
| ABALONE | 2923 | 584 | 670 | 71 |
| DELTA-AIRLOINS | 4990 | 998 | 1141 | 55 |
| ELEVATORS | 11619 | 2323 | 2657 | 182 |
| INSURANCE | 936 | 187 | 215 | 36 |
| MNIST | 48000 | 12000 | 10000 | 784 |
| FASHION | 48000 | 12000 | 10000 | 784 |

## B  Datasets

We employed the following datasets for our empirical evaluation, taken from the UCI Machine Learning repository and other regression [11] or classification suites.

**Description of the datasets**  ABALONE[7] [19] contains several physical measurements on abalone specimens used to predict their age. DELTA-AIRLOINS collects mechanical measurements for the task of controlling the ailerons of a F16 aircraft while the task is to predict the variation of the action on the ailerons. ELEVATORS comprises measurements also concerned with the task of controlling a F16 aircraft (different from DELTA-AIRLOINS), although the target variable here refers on controlling on the elevators of the aircraft. In INSURANCE[8] one wants to predict individual medical costs billed by health insurance given several personal data of a patient. MNIST [9] comprises gray-scale handwritten digit images used for multi-class classification. FASHION [10] is a 10-class image classification challenge concerning fashion apparel items.

**Preprocessing steps**  We preserve for all dataset their train and test splits if present in their respective repositories, or create a new test set comprising 20% of the whole data. Moreover we reserve a 10% portion of the training set as validation data used to monitor (parameter and/or structure) learning of our models and perform early-stopping.

We perform discretization of the continuous features in the regression datasets as follows. We first try to automatically detect the optimal number of (irregular) bins through adaptive binning by employing a penalized likelihood scheme as in [22]. If the number of the bins found in this way exceeds ten, we employ an equal-width binning scheme capping the bin number to ten, instead. Once the data is discrete, we encode them as binary through the common one-hot encoding, to accommodate the requirements of the PSDD learner we employed [16].

For image data, we binarize each sample by considering each pixel in it to be 1 if its original value exceed the mean value of that pixel as computed on the training set.

Statistics for all the datasets after preprocessing can be found in Table 1.

**Runtime of the algorithms**  In table 2, we report the runtimes for our method versus MICE and Monte Carlo method, which approximates the expected prediction by sampling the generative model and evaluating on the discriminative model. As we see, our method's advantage in speed becomes more clear we go to bigger models. The runtime of the Monte Carlo algorithm depends on number of samples and the size of the generative circuit, however runtime of MICE also depends on missing percentage of features and grows as more features go missing. Due to this, and the fact that our method was doing better, we stopped the MICE experiments early at 30% missing for the ELEVATORS dataset.

Table 2: Statistics on the runtime of our algorithm versus MICE and the Monte Carlo Sampling algorithm. The reported times for prediction times are for one configuration of the experiment. As we tried 10 different missingness percentages and repeated each 10 times, the total time of experiment is 100 times the value in the table. Learning is done only once.

| | **Time (seconds)** | | | |
| | **ours** (learning) | **ours** (prediction) | **MICE** | **MC** |
|---|---|---|---|---|
| ABALONE | 82 | 20 | 43 | 117 |
| DELTA | 53 | 24 | 27 | 126 |
| INSURANCE | 40 | 13 | 11 | 20 |
| ELEVATORS | 2105 | 31 | 364 | 994 |

**Computing Infrastructure**   The experiments were run on a combination of a server with 40 CPU cores and 500 GB of RAM, and a laptop with 6 CPU cores and 16 GB of RAM. The server was mainly utilized for learning the circuits, albeit not using all the memory, and to parallelize different runs of the missing value experiments. No GPUs were used for the experiments as probabilistic circuit libraries do not support them yet.

To report the runtimes in table 2, we did a separate run of each method on the same machine (the laptop) for fair comparison of the runtimes.

## Footnotes

[7]https://archive.ics.uci.edu/ml/datasets/abalone

[8]https://www.kaggle.com/mirichoi0218/insurance

[9]http://yann.lecun.com/exdb/mnist/

[10]https://github.com/zalandoresearch/fashion-mnist