[Reviews · NeurIPS 2019]

Reviewer 1



Good quality research, but I think the potential significance and impact of the results is low. The work has limited application, beyond simple toy models and does not provide new significant theoretical insights.

Reviewer 2



Originality: This work is an application of the much explored idea (partition functions, belief propagation, dynamic programing, etc.) that sums/max operations on functions whose variable dependencies decompose into a tree-like structure and can be carried efficiently via recursive procedures and caching. The paper makes a small contribution over the work of Guy Van den Broeck. The quality of the work is solid. The proofs, in particular, are very clean, and mostly amount to rearranging multiple summations of products. Definitions and statements are precise. Great job! The clarity of the writing and explanations is very good in general, but the paper is, at times, too concise in its explanation, making it easy to read only for people previously exposed to these topics. From the theoretical point of view, the significance of work is modest, since the results are not surprising, and are of similar flavor as ideas regarding how to efficiently compute sums/products of functions. The push to bring circuit theory, from the theory of computation field, to ML is laudable, and important. However, from a practical point of view, most technology seems to be driven by continuous and inexact methods (training NNs, etc.), not by discrete and exact methods. From a practical point of view, the significance is unclear. The data-sets used are very small and simple. Also, it would be good if the authors compared their technique with methods other than imputation methods, and small data-sets. Maybe compare against some sampling methods to estimate the moments.

Reviewer 3



The paper considers an important problem that has also been mentioned in the literature. The algorithms proposed are non-trivial and the paper is overall very well-written and gives a clear exposition of the concepts and the ideas used throughout with many intuitive discussions which were useful. However, some parts could be improved. Specifically, I suggest adding a related work section wherein more details about the previous work and their relation to the current work are discussed. This would also help non-experts learn more about the true contribution of this paper and the concepts introduced. Nonetheless, I believe the considered problem could of interest to many people at NeurIPS and would like to see the paper accepted if there is room. Minor comment === Line 29: an arbitrary discriminative models --> model?

Reviewer 4



Logical circuits, are DAG structures used to represent functions. Specific structural properties of these DAG structures and the corresponding logic circuit representations are used to demonstrate ways of efficient computation of higher order expectations. If the logic circuit representations of the probabilities and discriminative function satisdy specific properties, these moments are efficiently computed. The proofs of propositions provided in Section 4 (1 & 2) seem to reasonably check out. However that of Theorem 2 & 3 need more elaborate description that seems missing. Overall the paper seems to be extending results of [10]. The results are an interesting application of very old methods (logical circuit representations) to the problem of calculating expectations. The paper is clearly written and organized well barring minor typos. I am not really convinced of the significance of this work given the negative results provided in Theorems 2&3, which suggest that seemingly for simple classes of discriminative functions calculating moments using this framework is #P-hard.

Reviewer 5



The manuscript considers basic statistical questions regarding reasoning about the expected outcome of a predictive model. Efficiently computing even the expectation (first moment) is a known challenge even for simple predictive models and simple generative models (e.g. logistic regression and naive Bayes distribution). The authors give a pair of generative and discriminative models (family of structured probabilistic circuits) that enables tractable computation of expectations (and higher order moments as well), in some cases approximately, b) provide algorithms for computing moments of predictions wrt generative models and c) show that the utility of the algorithms in handling missing data during prediction time compared to standard imputation techniques on some datasets. The paper is organized and written well, there are some good technical contributions. But I'm unable to get a good grasp on the overall significance and merit of this work - partly because the authors aren't convincing enough throughout the paper. I'm also not entirely sure if NeurIPS readers are the right audience for this work - not just in terms of applying these results in practice, but primarily in terms of taking the scope of this work forward. In the problem set up of Section 2, I wasn't entirely sure where the paper was heading towards - for example, why should one be interested in *exact* computation of mean/moments at all, given that the predictive models are already constrained by the availability of training data? Furtheremore, why is this a new problem in ML/computer science? I'd think these are fundamental questions in statistics and inter-disciplinary computational science communities (I don't see references to these in Sec 2). I also didn't see an exposition of jump from logical circuits to real-valued data/inferences anywhere in the paper (nor at least a hint in the main Section). The paper does cite references to existing work that deal with these issues, but the gist of idea needs to be conveyed in Section 3 to make the ideas grounded and concrete. In fact, the Section is written assuming PSDD is a "given" which is already concerning (of course, the reader might guess that it will be learned from data as well). Not until later in the experiments does the paper give a concrete footing - "Our advantage clearly comes from the PSDD learning a better density estimation of the data distribution, instead of having fixed prior assumptions about the features." I liked the experimental design and results, but I don't know why the authors don't talk about nor compare to simple density estimation or other statistical tools to compute expectations of (non-linear) predictive functions from training data (even without caring about characterizing the data distribution, i.e.). --- I've read the authors' response; my stance on the paper is however unswayed. Overall, I like this work for some of the technical contributions, but what I'm still not clear about is why accurately determining the mean/moments is significant from machine learning perspective, as well as lack of comparisons/references to work from Stats community.

[Author Response · NeurIPS 2019]

|  | RMSE | | | | | | Time (seconds) | | | |
|---|---|---|---|---|---|---|---|---|---|---|
|  | **ours** (20%) | **ours** (80%) | **MICE** (20%) | **MICE** (80%) | **MC** (20%) | **MC** (80%) | **ours** (learning) | **ours** (prediction) | **MICE** | **MC** |
| Abalone($1e0$) | $2.72_{\pm.04}$ | $2.99_{\pm.02}$ | $2.71_{\pm.07}$ | $4.50_{\pm.09}$ | $2.80_{\pm.04}$ | $3.39_{\pm.07}$ | 82 | 20 | 43 | 117 |
| Delta ($1e-4$) | $1.84_{\pm.02}$ | $2.61_{\pm.05}$ | $1.88_{\pm.02}$ | $3.27_{\pm.06}$ | $1.91_{\pm.02}$ | $2.80_{\pm.04}$ | 53 | 24 | 27 | 126 |
| Insurance($1e3$) | $5.84_{\pm.24}$ | $10.1_{\pm.43}$ | $6.24_{\pm.38}$ | $13.0_{\pm.77}$ | $6.14_{\pm.28}$ | $10.1_{\pm.49}$ | 40 | 13 | 11 | 20 |
| Elevators($1e-2$) | $0.46_{\pm.01}$ | $0.64_{\pm.01}$ | $0.44_{\pm.01}$ | $1.15_{\pm.04}$ | $0.55_{\pm.01}$ | $0.83_{\pm.01}$ | 2105 | 31 | 364 | 994 |

We thank the reviewers for their valuable feedback. We appreciate they recognize that the paper is "*well-written*"
"*clear*" (R#2, R#3, R#4, R#5), whose technical contribution "*quality is solid*" (R#2), "*very good*" (R#1, R#5) and
"*non-trivial*" (R#3) while it considers "*an important problem in ML*" (R#3, R#4) which can "*be of interest to many
people at NeurIPS*" (R#3). We hope to address all questions and concerns raised in the following.

**[Reviewer #1]** **1. Limited impact.** We disagree with the reviewer. As also noted by R3 and R4, computing the expected
predictions of a model lies *at the core of ML and statistics*. Among the plethora of ML problems that would benefit
from our algorithm, there are: missing value imputation, feature selection, several formulations of fairness as well as
computing integral probability metrics, i.e., a fundamental way to assess the distance between distributions (e.g., see
the popular Wasserstein distance). In this paper, we tackled just the first one in the list to show the effectiveness of
our algorithm. We are actively working on applying it to the other application scenarios. **2. Toy models.** the structural
properties we require for our circuits *do not compromise expressiveness*: PSDDs are SOTA density estimators that are
comparable to MADEs and VAEs on many benchmarks (compare the results in [1] w.r.t. those in [2]) and LCs are able
to achieve the same accuracy of much more complex neural networks (e.g., Resnets cfr. [3]).

**[Reviewer #2]** **1. Results easily follow from literature.** Our technical contribution goes beyond the results known in
the literature of circuits. Classic sum/max problems only require simpler structural properties and they focus on one
circuit at a time. E.g., sums (marginals) require only decomposability and smoothness, with the addition of determinism
for max problems (MAP). Here, for expectations, we need to deal with a pair of circuits and we require them to be both
structured decomposable and to share the same vtree. We agree that computations are simple, i.e., elegant, *once the
aforementioned requirements have been elicited*. Eliciting them, however, is definitely non-trivial and has not been
explored in the literature so far for expectations. Indeed, our work has been made possible only very recently, after
discriminative circuits satisfying such structural properties have been introduced in [2]. **2. Simple datasets.** Statistics
are reported in the Appendix. Note that our contribution is more theoretical than empirical. As such, our experiments are
meant to showcase the (theoretically expected) effectiveness of our algorithms when a reasonably accurate generative
model is available, across different real world datasets. Our circuits are expressive enough to model larger datasets
(see our answer to R#1.2) and learning them would scale: in many cases it is easier to learn a LC than a neural net
(e.g., see [3]). **3. Approximate inference alternatives.** Whenever we are able to compute expectations exactly for
regression (Thm 1), we might want to consider approximations only to speed computations. This is however not
necessary in practice, as our algorithm is very efficient due to caching (see next point). For classification, we resort to
approximations but, unfortunately, we cannot provide anytime guarantees. We will discuss and cite related works on
anytime approximations as it is a sensible venue to explore. **4. Run times.** We report in the top table the RMSE and the
avg. time to predict one test sample for regression with 20% and 80% missing values (we will report all results in the
paper) and compare to Monte Carlo (MC) estimates via 200 samples drawn from the PSDD. Our method is not only
faster but more accurate than MC (and MICE). Note that the time to learn the regression circuit is easily amortized after
the prediction of a few samples. **5. Code and figures.** We will make the figures and code more accessible.

**[Reviewer #3]** **Related works.** We will add a detailed discussion of previous approaches to computing moments, such
as Monte Carlo methods (along with experiments; see response R#2.4) and missing value imputation techniques.

**[Reviewer #4]** **1. Proofs.** We will provide more detailed proofs for Thms 2 & 3. Specifically, we will show in detail
how we can reduced our case to those whose complexity has been previously derived. **2. Extension.** The work in [4]
avoids computing expectations by distilling a (simple) generative model from a (simple) discriminative model. We take
another path, which is not a trivial derivation. See also our answer to R#2.1. **3. Negative results.** For regression (Thm
2), the needed structural constraints do not hinder expressiveness. See our answer to R#1.2. For classification (Thm 3),
we need to resort to approximations (which are still more effective than competitors for missing values). Note that Thm
3 does not state that there cannot exist a circuit pair with additional structural assumptions enabling exact computations.

**[Reviewer #5]** **1. Wrong audience.** Our method can be impactful to many ML scenarios (see our answer to R#1.1). As
R#3 and R#4 recognize, NeurIPS is a sensible venue. **2. Finite data.** We exploit a generative model as a proxy to the
true data distribution. Indeed, we learn it from data, and the better density estimator it is, the more accurate the expected
predictions will be. We will discuss this in Section 3 along with how to deal with continuous data. **3. Baselines.** We
will add the comparison with MC estimates over samples from the same PSDD (see our answer to R#2.4).

**[References]**[1] Liang et al. "Learning the Structure of Probabilistic Sentential Decision Diagrams" UAI 2017 **[2]** Peharz et al.
"Random Sum-Product Networks: A Simple and Effective Approach to Probabilistic Deep Learning" UAI 2019 **[3]** Liang et al.
"Learning logistic circuits" AAAI 2019 **[4]** Khosravi et al. "What to expect of classifiers?" IJCAI 2019


[Meta-Review · NeurIPS 2019]

This manuscript proposes novel approaches for efficiently computing expectations with respect to generative models under certain assumptions. The reviewers discuss the clarity and novelty of the main results as a strength of the manuscript. On the other hand, the reviewers point out weaknesses related to the strength of the assumptions and the extent to which they may hold on real data, and may have wide applicability. Reviewers are concerned about sufficient empirical evaluation, perhaps exploring some real-world settings where the proposed approach may be relevant. There are also issues with the conciseness of the writing in some parts. The authors are encouraged to carefully incorporate the feedback from reviewers in subsequent versions.